# Competition and evolutionary selection among core regulatory motifs in gene expression control

Andras Gyorgy ⬥[1] ✉

Gene products that are beneficial in one environment may become burdensome in another, prompting the emergence of diverse regulatory schemes that carry their own bioenergetic cost. By ensuring that regulators are only expressed when needed, we demonstrate that autoregulation generally offers an advantage in an environment combining mutation and time-varying selection. Whether positive or negative feedback emerges as dominant depends primarily on the demand for the target gene product, typically to ensure that the detrimental impact of inevitable mutations is minimized. While self-repression of the regulator curbs the spread of these loss-of-function mutations, self-activation instead facilitates their propagation. By analyzing the transcription network of multiple model organisms, we reveal that reduced bioenergetic cost may contribute to the preferential selection of autoregulation among transcription factors. Our results not only uncover how seemingly equivalent regulatory motifs have fundamentally different impact on population structure, growth dynamics, and evolutionary outcomes, but they can also be leveraged to promote the design of evolutionarily robust synthetic gene circuits.

Much effort has been devoted to uncovering organizing principles of living cells in systems biology, and also to devising design guidelines to ensure the predictable behavior of cellular dynamics in synthetic biology[1–3]. As a result, we not only better understand the processes underpinning bacterial chemotaxis perfected by evolution[4], but also how to implement integral feedback to ensure robust perfect adaptation[5–8]. These results are generally made possible by interpreting complex dynamical systems as a collection of core components wired together, each realizing a well-defined and highly-optimized information processing function[9–11]. While this view offers a powerful reductionist approach to design and analyze networks of daunting complexity, recent results also highlight its limits as recurring motifs can exhibit a wide range of dynamical responses depending on their biophysical parameters and context[12–21].

Among common network motifs, activation and repression are the most fundamental building blocks. They are functionally equivalent (Fig. 1a), as the expression of a gene product can be regulated by relying on an inducer that either activates an activator (positive control), or relieves the repression of a repressor (negative control). Savageau proposed that according to the use-it-or-lose-it principle, positive/negative control emerges when gene products are often/rarely needed[22–24], ensuring that cognate binding sites are occupied by the transcription factors (TFs) most of the time, thus minimizing the probability of fitness-reducing errors[25]. Conversely, as a mutated regulator represents a fitness cost only when it is needed, the wear-and-tear principle suggests that it may be evolutionary advantageous to instead minimize the usage of regulators to reduce the negative impact of eventually inevitable mutations[26], motivated by the well-established population genetics concept of genetic robustness[27–30].

While precise temporal control of a beneficial gene product may result in an advantage, the expression of the required regulator carries its own bioenergetic cost[31]. Crucially, for both positive and negative control, the regulator is only required when the inducer is present, otherwise its expression is gratuitous. Therefore, it may be

[1]Division of Engineering, New York University Abu Dhabi, Abu Dhabi, UAE. ✉e-mail: andras.gyorgy@nyu.edu

advantageous to have the regulator under autoregulatory control to ensure it is expressed only when needed (Fig. 1b). Understanding the competition and evolutionary selection among the core regulatory motifs in Fig. 1 could shed light on organizing principles of living organisms as similar just-in-time regulation is a wide-spread feature in natural systems[32–36], as well as guide the design of synthetic gene circuits when selecting among alternative modes of regulation[37–52].

Motivated by the central role of autoregulation in systems and synthetic biology, we characterize how demand for a beneficial gene product, mutation rate of its regulator, population size, selection pressure, regulatory delay, and the timescale of environmental shifts together determine the optimal choice among the motifs in Fig. 1. We show that (i) autoregulation generally dominates, (ii) the dominant strategy typically agrees with the wear-and-tear principle, and (iii) while self-repression of the regulator curbs the spread of loss-of-function mutations, self-activation instead facilitates their

propagation. We further demonstrate that the reduced bioenergetic cost of autoregulation may contribute to its ubiquitous nature in gene regulatory networks, and how our work could aid the design of evolutionarily robust synthetic gene circuits.

## Results

### Mathematical model

Building on a quantitative framework[26] inspired by demand theory[22–24], the mathematical model underpinning our analysis comprises the changing environment, random mutations, and fitness-based selection. Typical values of the model parameters are discussed in the "Methods".

We consider an evolutionary scenario where cells are exposed to environmental variations. This is modeled via the concentration of the inducer I that varies periodically between high and low values (Fig. 2a). Within each period $T$, these correspond to the induced and non-induced

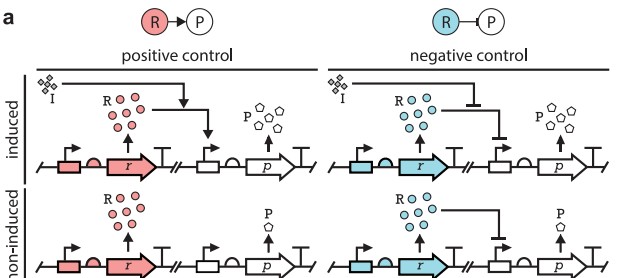
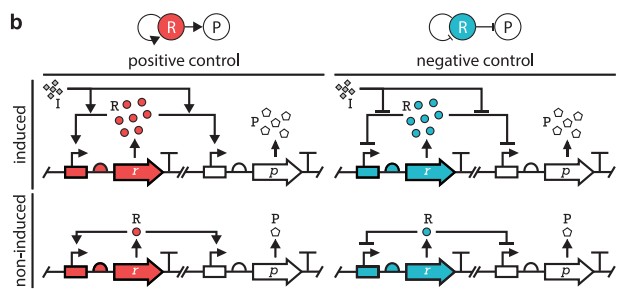

**Fig. 1 | Functionally equivalent core regulatory motifs in gene expression control.** The regulator R and the product are expressed from their genes r and p, respectively. Gray squares denote the inducer I. In the absence of the inducer, the product (and the regulator for autoregulated motifs) may be synthesized at a low

basal level which is negligible compared to the concentration in the induced state (Supplementary Section 1). The motifs may be embedded in complex regulatory pathways (Supplementary Fig. 3). **a** Expression of R is non-autoregulated. **b** Expression of R is autoregulated.

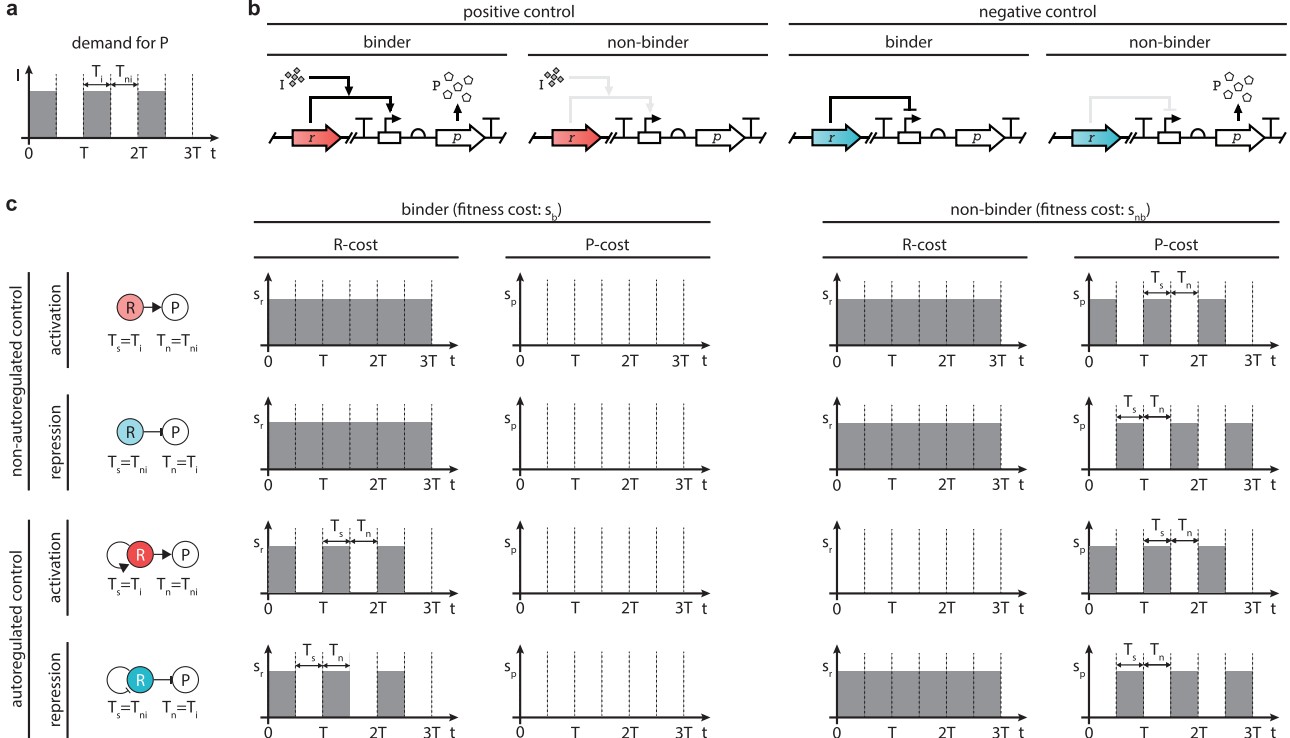

**Fig. 2 | The evolutionary setting combines environmental shifts, mutations, and fitness-based selection. a** Each period $T$ comprises an induced phase $T_i$ (P should be expressed) and a non-induced phase $T_{ni}$ (P should not be expressed). **b** Non-binders fail to synthesize the product despite the presence of the inducer I in

case of positive control, and conversely, they continuously produce it in case of negative control. Potential autoregulation of R is not shown. **c** Overall fitness depends on the sum of the P-cost and the R-cost.

phases lasting $T_i$ and $T_{ni}$, respectively. During the former, expression of a gene product P confers a fitness advantage, whereas during the latter its unnecessary synthesis a fitness cost. The fraction $D = T_i/T$ hence measures the demand for the beneficial gene product.

In this evolutionary setting, loss-of-function mutations that affect the regulator R result in non-functional variants. The emergence of these non-binders occurs at rate $v_-$ from a functional binder, whereas gain-of-function mutations happen at rate $v_+$ (Fig. 2b). We assume that these mutation rates are constant and independent of the mode of gene regulation. We consider the population size $N$ to remain constant over time, and denote the size of the binder and non-binder sub-populations with $N_b(t)$ and $N_{nb}(t)$, respectively.

Finally, each period alternates between selection and neutral phases lasting $T_s$ and $T_n$, respectively (Fig. 2c). During the former, non-binders suffer the cost $s_p > 0$ due to either the presence of P during $T_{ni}$ for negative control, or its absence during $T_i$ for positive control. Additionally, cells incur the expense $s_r > 0$ when the regulator R is synthesized. Selection pressure hence stems from two sources: expression of P not matching the environmental condition (P-cost), and expression of R (R-cost). For non-autoregulated control, binders and non-binders suffer identical R-cost throughout the entirety of each period, thus selection against non-binders occurs solely based on their non-zero P-cost. Autoregulation of R reduces the time when the R-cost is suffered (Fig. 2c), thus holding the potential to provide an evolutionary advantage over non-autoregulated control (Supplementary Figs. 1 and 2). Therefore, we next quantify the average fitness cost of each core regulatory motif in Fig. 1 to compare their performance.

## Fitness cost in large populations

We first quantify the performance of the control schemes in large populations, when sampling fluctuations are negligible. The evolution of the fraction $x = N_{nb}/N$ of non-binders in the population is governed by the deterministic dynamics (Supplementary Section 2)

$$\dot{x} = \nu_- - x(\nu_+ + \nu_- + s) + x^2 s, \tag{1}$$

where $s(t) = s_{nb}(t) - s_b(t)$ is the relative selection pressure against non-binders, with $s_b(t)$ and $s_{nb}(t)$ denoting the fitness cost encountered by the binders and non-binders, respectively (Fig. 2c). Thus, over each period the average fitness cost $\bar{s}$ is

$$\bar{s} = \frac{1}{T} \int_0^T \left[ x(t) s_{nb}(t) + (1 - x(t)) s_b(t) \right] dt. \tag{2}$$

Considering only non-autoregulated control, activation/repression dominates at low/high demand for weak selection (Fig. 3a). For intermediate values of the demand (Supplementary Figs. 4–6), the two control schemes offer comparable performance for short periods and when control is expensive ($s_r \approx s_p$). Selection pressure further amplifies this effect (Supplementary Figs. 7–9), and we recover the results presented in ref. 26: when the fraction of non-binders does not appreciably change as a result of frequent environmental shifts (Supplementary Section 1), positive and negative control perform similarly, otherwise activation/repression dominates for low/high demand, matching the wear-and-tear principle.

When considering all motifs in Fig. 1, non-autoregulated control is generally replaced by its autoregulated counterpart as the dominant strategy (Fig. 3b). However, two major differences do emerge. First, for low demand, while non-autoregulated activation and repression often have comparable performance (especially for strong selection, Fig. 3a), the parameter region where self-activation emerges as a clear winner expands significantly due to the elimination of the R-cost for non-binders (Fig. 2c). Second, when demand is high, control is expensive ($s_r \approx s_p$), and the combined impact of mutations and selection is weak (Supplementary Section 1), self-activation emerges as dominant (red star in Fig. 3b). To understand this, consider the case when $s_r = s_p = s_0$, yielding the fitness cost $Ds_0$ for self-activation (Fig. 2) and $Ds_0 + 2x_0(1 - D) > Ds_0$ for self-repression where $x_0$ is the approximately constant value of $x$ throughout the period (Supplementary Section 2). Thus, while the winner emerges according to the wear-and-tear principle when only non-autoregulated control is considered (Fig. 3a), the dominant strategy may be aligned with the use-it-or-lose-it principle in

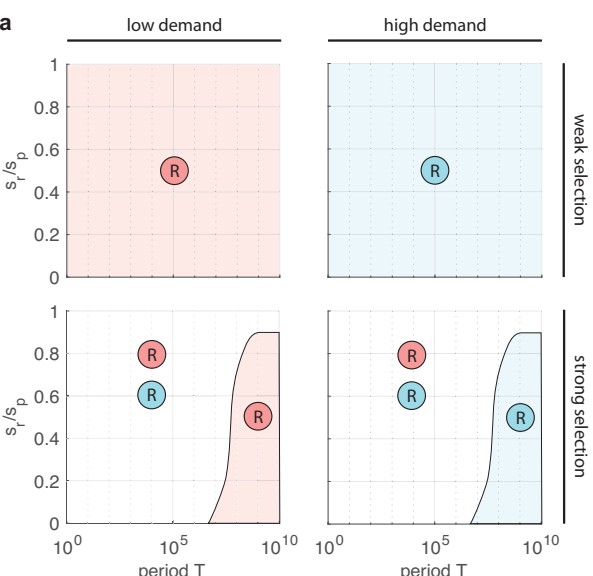

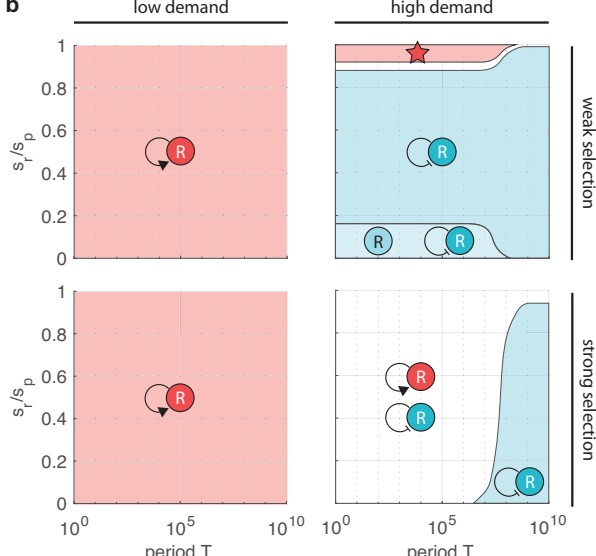

**Fig. 3 | Evolutionary advantageous regulatory motifs in large populations.** Low and high demand corresponds to $D = 0.05$ and $D = 0.95$, respectively. In all plots $v_- = 10^{-7}$, $v_+ = v_-/10$, weak and strong selection indicates $s_p = 10v_-$ and $s_p = 100v_-$, respectively. Colors indicate the control scheme(s) with the lowest fitness cost $\bar{s}$: in case of a single winner, the second best is dominated by at least 1%; in case of multiple winners, their difference is less than 1% and all other variants are at least 1% worse. Star indicates where the unique dominant strategy matches the use-it-or-

lose-it principle. For detailed simulation data, see Supplementary Figs. 4–15. **a** Only non-autoregulated control is considered. In the light red/blue regions activation/ repression dominates, in the white regions they offer comparable performance. **b** Both non-autoregulated and autoregulated control are considered. Dominant strategies: self-activation (dark red), self-repression (dark blue), non-autoregulated and autoregulated repression (light blue), self-activation and self-repression (white).

the presence of autoregulation (Fig. 3b). The region where this occurs shrinks with increasing demand (Supplementary Figs. 10–15).

## Fitness cost in small populations

The impact of sampling fluctuations (genetic drift) becomes more pronounced as the population size decreases, thus we next characterize performance in the presence of stochastic effects. To this end, we first consider the standard Wright-Fisher model with constant population size $N$[53,54], then its diffusion approximation[55] as it significantly accelerates the computation of the fitness cost without compromising accuracy (Supplementary Fig. 16).

Focusing on non-autoregulated control first, it was previously reported that while the wear-and-tear principle dominates in large populations, it is replaced by the use-it-or-lose-it principle as $N$ decreases[26]. Although this can happen, it only occurs when selection pressure is strong ($s_p \gg v_-$), and there is a reversal as the population size further decreases (Fig. 4a), which is not discussed in ref. 26. The source of these two transitions is the varying frequency of non-binders occasionally taking over during neutral periods due to sampling fluctuations (unlike in large populations). The prevalence of such events is inversely proportional to the population size, the duration of the

selection phase, and the selection pressure, and they carry a significant penalty as binders only slowly re-emerge as a result of rare gain-of-function mutations.

To better understand this phenomenon, consider the low demand case (the high demand case can be analyzed similarly, with positive and negative regulation swapped). As population size starts to decrease, non-binders take over more frequently for activation than for repression due to the shorter selection period (Supplementary Fig. 18), giving rise to the region where the dominant strategy is consistent with the use-it-or-lose-it principle (blue star in Fig. 4a). A similar shift occurs for negative control as well, only at lower population size due to the longer selection phase, and the corresponding substantial fitness cost increase is what drives the reversal to positive control re-emerging as the dominant strategy, this time at the small population limit (Fig. 4a). These transitions happen only when selection pressure is sufficiently strong to ensure that non-binders are eliminated in large populations.

When considering all core motifs featured in Fig. 1, the picture that emerges in Fig. 4b is qualitatively similar when environmental shifts happen frequently ($T \ll 1/v_-$) and when they occur rarely ($T \gg 1/v_-$). For low demand, the trend echoes our previous findings in

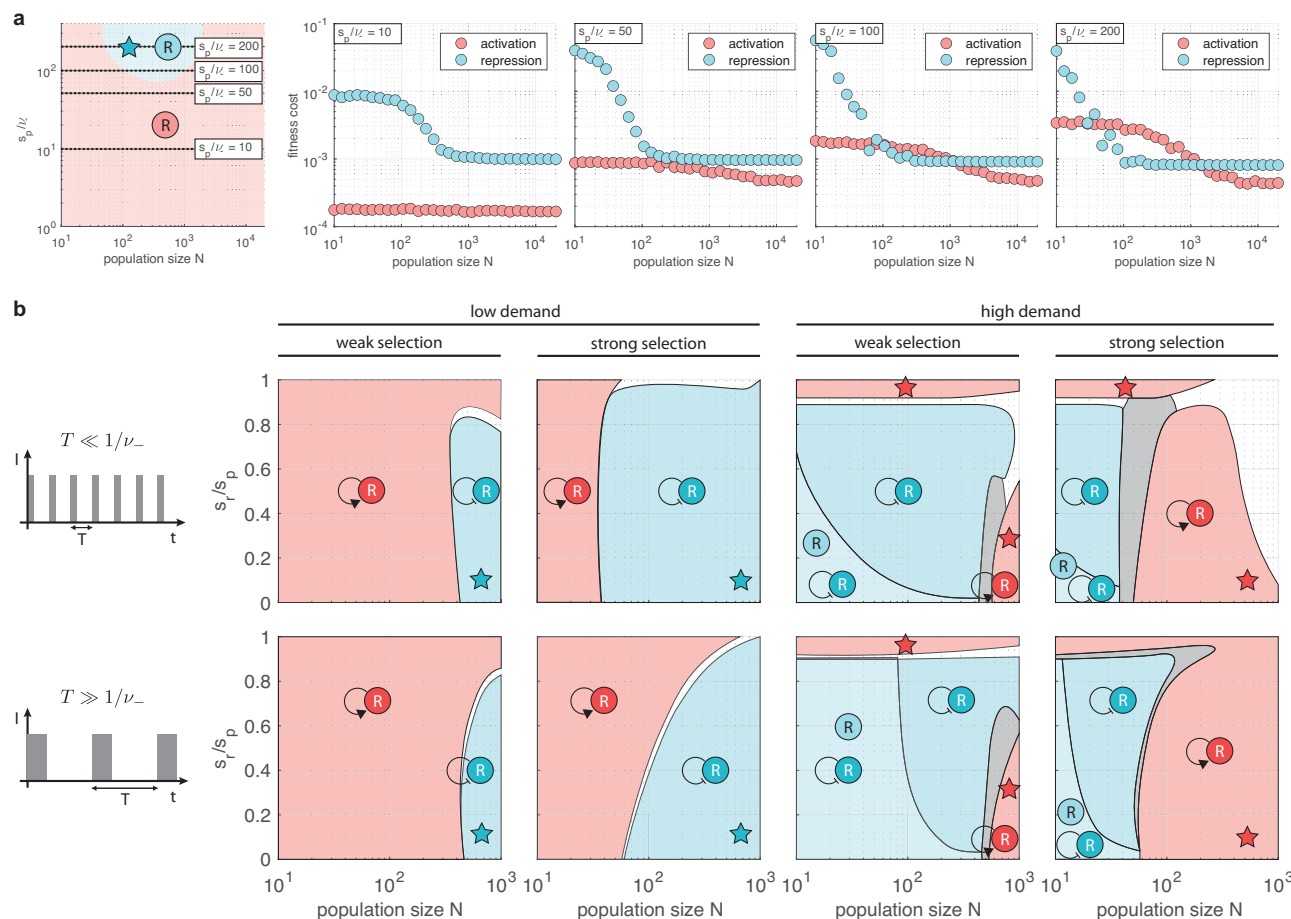

**Fig. 4 | Evolutionary advantageous regulatory motifs in small populations.** Colors indicate the control scheme(s) with the lowest fitness cost $\bar{s}$: in case of a single winner, the second best is dominated by at least 1%; in case of multiple winners, their difference is less than 1% and all other variants are at least 1% worse. Star indicates where the unique dominant strategy matches the use-it-or-lose-it principle. **a** Dominant strategy considering only non-autoregulated control. Simulation parameters are $v_- = 10^{-3}$, $v_+ = v_-/10$, $T_i = 100$, $T_{ni} = 5000$. For detailed simulation data, see Supplementary Fig. 17. Light red/blue indicates the regions where activation/repression dominates. Data obtained using the stochastic sampling algorithm described in the "Methods". **b** In all plots $v_- = 10^{-3}$, $v_+ = v_-/10$, weak

and strong selection indicates $s_p = 10v_-$ and $s_p = 100v_-$, respectively. Low and high demand corresponds to $D = 0.05$ and $D = 0.95$, respectively, together with $T = 10^2$ and $T = 10^4$ for when $T \ll 1/v_-$ and $T \gg 1/v_-$, respectively. For detailed simulation data, see Supplementary Fig. 21. Dark red/blue indicates regions where self-activation/self-repression dominates, light blue corresponds to regions where non-autoregulated and autoregulated repression dominate together, in the white regions autoregulated activation and repression have comparable performance dominating their non-autoregulated counterparts, gray encompasses all other outcomes. Data obtained using the diffusion approximation described in the "Methods".

Fig. 4a when considering only non-autoregulated control. The region where the dominant strategy is underpinned by the use-it-or-lose-it principle expands with selection pressure, and it is sandwiched between population sizes where it is replaced by the wear-and-tear principle from both below (Fig. 4b) and above (Fig. 3b), only this time self-activation and self-repression dominate instead of their non-autoregulated counterparts due to their reduced R-cost (Fig. 2c). While in the high demand case the situation is similar (with self-activation and self-repression swapped), one crucial difference does emerge: in the small population limit non-autoregulated and autoregulated repression have comparable performance. To understand this, note that unlike for activation where autoregulation decreases the fitness cost by $s_r$ throughout the entire period, for repression there is no reduction in case of non-binders (Fig. 2c), yielding comparable performance for non-autoregulated and autoregulated control. This effect becomes more pronounced with decreasing selection pressure (Fig. 4b), which increases the probability of non-binders taking over the population due to stochastic fluctuations (Supplementary Figs. 19 and 20).

In summary, the presence of sampling fluctuations largely preserves the wear-and-tear principle behind the dominant strategy considering both non-autoregulated and autoregulated control. The use-it-or-lose-it principle emerges only in a narrow slice of the parameter space (which further decreases with selection pressure), for instance, within a confined range of the population size. Thus, in addition to providing a more complete picture about the competition between the two non-autoregulated motifs by revealing the reversal to the wear-and-tear principle in the small population limit, our results also significantly expand prior work[26] by comparing the performance of all four core regulatory motifs featured in Fig. 1 in the presence of genetic drift.

## Autoregulation can result in unwanted selection pressure

By eliminating the gratuitous expression of the regulator during the non-induced phase (Fig. 2c), autoregulation generally outperforms its non-autoregulated counterpart for both activation and repression (Figs. 3 and 4). For positive and negative control, however, autoregulation has drastically different impact on the population composition, as well as on how rapidly it changes.

To illustrate the differential impact of autoregulation on the fraction of non-binders $x$, consider first non-autoregulated control. Increasing $s_r$ represents an additional and identical fitness cost for both binders and non-binders (Fig. 2c), hence the population-level composition remains unaffected, since the evolution of $x$ according to (1) depends on the difference $s = s_{nb} - s_b$. Conversely, while autoregulation yields $s = 0$ during the neutral phase, it results in $s = s_w$ with $s_w = s_p - s_r < s_p$ and $s_w = s_p + s_r > s_p$ during selection for self-activation and self-repression, respectively (Fig. 5), compared to $s_w = s_p$ for non-autoregulated control. This change in selection pressure thus means (i) more stringent elimination of the non-binders when relying on self-repression, and (ii) the accumulation of loss-of-function mutations in case of self-activation (Fig. 5). The latter is especially concerning considering the fitness gain of self-activation relative to its non-autoregulated counterpart: should these two variants compete, the former would eventually take over the population, however, it could easily result in one dominated by non-binders (unlike in the case of self-repression where the R-cost instead promotes the elimination of deleterious mutations).

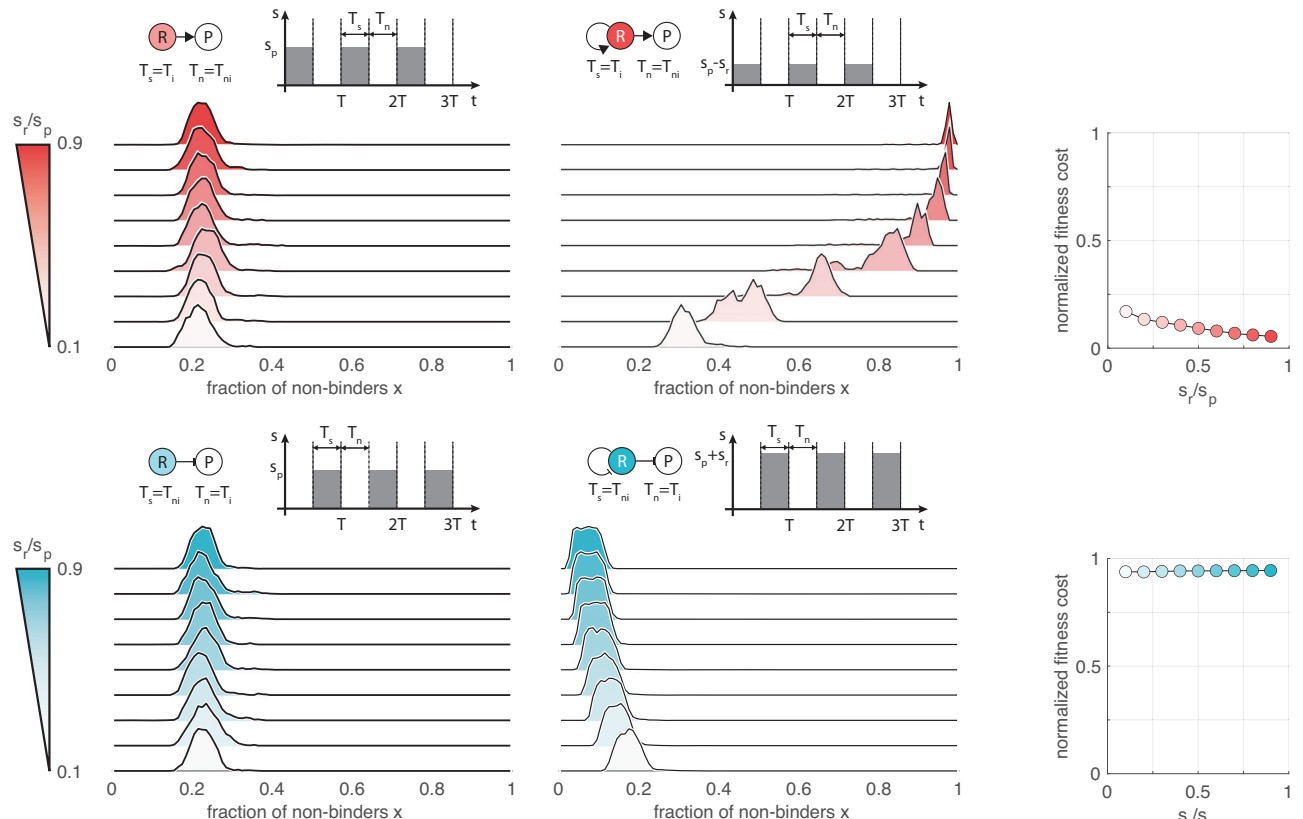

**Fig. 5 | Autoregulation can result in unwanted selection pressure.** In all plots $v_- = 10^{-3}$, $v_+ = v_-/10$, $s_p = 0.1$, $T = 100$, $N = 1000$, together with $D = 0.05$ in case of activation (red) and $D = 0.95$ in case of repression (blue). Darker shades indicate more expensive control (ranging from $s_r/s_p = 0.1$ to $s_r/s_p = 0.9$). Fitness cost is normalized to the baseline obtained when considering non-autoregulated control. Data obtained using the stochastic sampling algorithm described in the "Methods". Distributions are visualized with identical maximal heights. For detailed simulation data, see Supplementary Figs. 22 and 23.

Feedback also has a differential impact on how rapidly non-binders emerge and disappear in case of positive and negative autoregulation. To illustrate this, we define $T^* = \max(T_b, T_w)$ where $T_b = 1/\nu_-$ and $T_w = 1/(s_w + \nu_+)$ are the timescales for the build-up and wipe-out of non-binders due to mutations and selection. For short periods ($T \ll T^*$), the fraction of non-binders remains approximately constant throughout the entire period: non-binders are eliminated if $T_w < T_b$, otherwise they take over the population (Supplementary Fig. 24). For long periods ($T \gg T^*$), if $T_w < T_b$ then build-up of non-binders during $T_n$ is wiped out during $T_s$, otherwise non-binders persist throughout the entire period as loss-of-function mutations dominate the combined impact of selection and gain-of-function mutations during $T_s$ (Supplementary Fig. 24). Considering the typical range of model parameters ("Methods"), for all four regulatory schemes featured in Fig. 1 we have $T^* = T_b$ (Supplementary Section 4). Importantly, while $T_w$ decreases with $s_r$ in case of self-repression, autoregulation has the opposite impact for positive control, further hindering the elimination of non-binders from the population.

## Delay can cause non-autoregulated motifs to outperform autoregulation

For autoregulated motifs, the inducer also triggers the appearance and disappearance of the regulator. Since its synthesis and decay may take time, considerable delays could be introduced[56], resulting in an increased fitness cost when compared to the idealized scenario outlined in Fig. 2c, especially when feedback is realized in the form of regulatory cascades[57]. The negative impact of delay on the performance of autoregulation is illustrated in Fig. 6.

For short periods ($T \ll T^*$), both selection and mutation play a negligible role throughout each period, thus the fraction of non-binders $x$ remains approximately constant. Hence, in this case delay has no impact on the performance of autoregulated motifs: the dominant strategy in Fig. 6 remains unchanged when compared to the case without delay (Fig. 3b).

For long periods ($T \gg T^*$), the combined impact of mutation and selection can impact $x$, which may result in non-autoregulated control

outperforming autoregulation. For instance, in case of low demand, the fitness cost of non-autoregulated and autoregulated activation is approximately $s_r$ and $D s_p$, respectively (Supplementary Fig. 24). Therefore, while self-activation offers superior performance when regulation is expensive ($s_r \approx s_p$), non-autoregulated activation can dominate if control is instead affordable ($s_r \ll s_p$). These results hold for both weak and strong selection (Fig. 6), and also in the presence of stochastic fluctuations due to small population size (Supplementary Fig. 25). The high demand case can be analyzed similarly.

In the special case when $T \approx T^*$, autoregulation consistent with the use-it-or-lose-it principle emerges as dominant when selection pressure is sufficiently strong (stars in Fig. 6). To understand why this happens, here we focus on the low demand case when regulation is affordable ($s_r \ll s_p$), other scenarios can be analyzed similarly. For self-repression, non-binders are eliminated ($x \approx 0$) during the entire period (Supplementary Fig. 26) as a result of strong and long selection (due to low demand), yielding the average fitness cost $\bar{s} \approx D s_r$ (Fig. 2c). Conversely, for self-activation there is alternating build-up and wipe-out during $T_n$ and $T_s$ due to the shorter selection phase, resulting in $\bar{s} \approx \bar{x}_s D s_p$ where $\bar{x}_s$ denotes the average of $x$ during $T_s$ (Supplementary Fig. 26). Crucially, $\bar{x}_s$ increases with the delay, which is the driving force behind self-repression offering superior performance compared to self-activation, and eventually emerging as the dominant strategy for sufficiently strong selection (blue star in Fig. 6). Delay thus has the greatest impact in the special case when $T \approx T^*$ by triggering the emergence of regions where the dominant strategy is underpinned by the use-it-or-lose-it principle (marked by stars in Fig. 6). These regions expand with increasing selection pressure $s_p$, and they can appear even in the presence of brief delays (Supplementary Fig. 31).

## Feedback cost can cause non-autoregulated motifs to outperform autoregulation

In addition to introducing delay, autoregulation can also result in additional burden due to the increased expression of the regulator R that may be required to control its own expression[14,15]. To capture this, we next assume that while the R-cost for non-autoregulated control

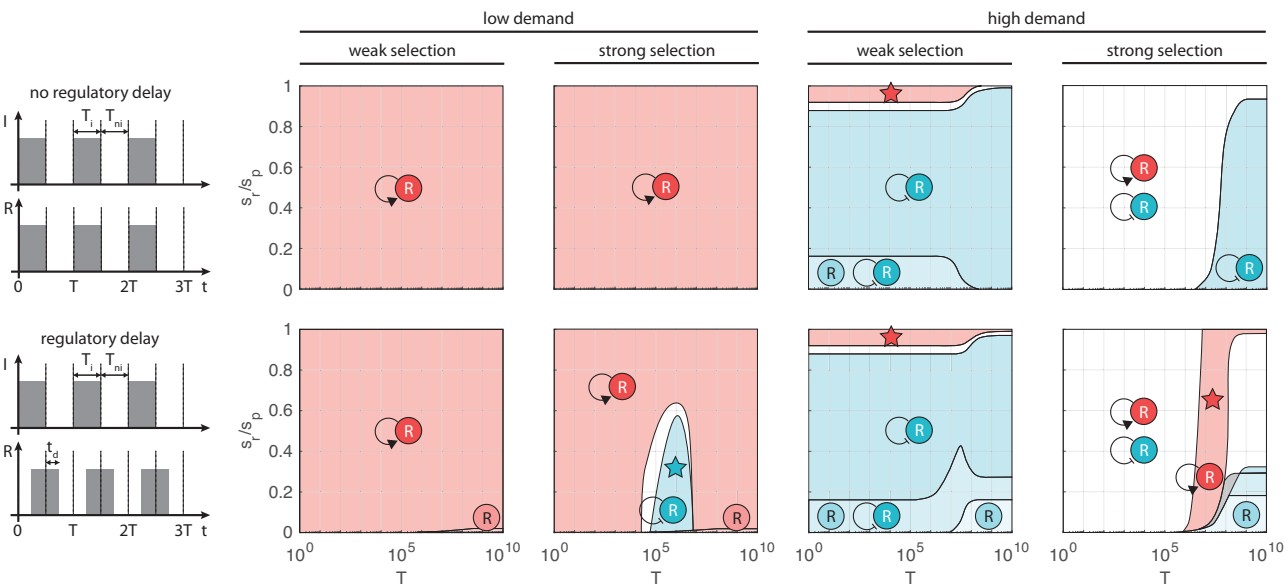

**Fig. 6 | Evolutionary advantageous motifs in the presence of delay.** In all plots $\nu_- = 10^{-7}$, $\nu_+ = \nu_-/10$. In case of autoregulation, the delay is $t_d = T/100$. Low and high demand refers to $D = 0.05$ and $D = 0.95$, weak and strong selection indicates $s_p = 10\nu_-$ and $s_p = 100\nu_-$, respectively. Colors indicate the control scheme(s) with the lowest fitness cost $\bar{s}$: self-activation (dark red); self-repression (dark blue); non-autoregulated and autoregulated repression (medium blue); non-autoregulated

repression (light blue); autoregulated activation and repression (white); all other outcomes (gray). For a single winner, the second best is dominated by at least 1%; in case of multiple winners, their difference is less than 1% and all other variants are at least 1% worse. Star indicates where the unique dominant strategy matches the use-it-or-lose-it principle. For detailed simulation data, see Supplementary Figs. 28–30.

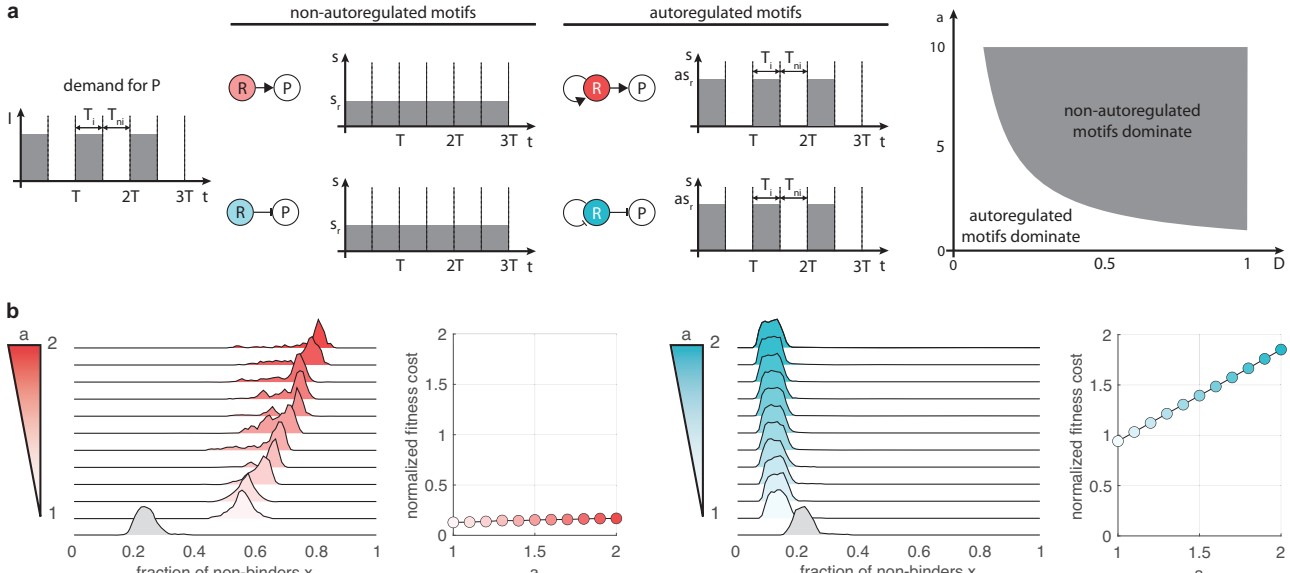

**Fig. 7 | Feedback cost has a differential impact on positive and negative control. a** Non-autoregulated variants outperform their autoregulated counterparts when $a > 1/D$ in the absence of mutations. **b** While feedback cost hinders the spread of loss-of-function mutations in case of self-repression, it instead facilitates their proliferation when relying on self-activation. Data obtained using the stochastic sampling algorithm described in the "Methods". In all plots $v_- = 10^{-3}$, $v_+ = v_-/10$,

$s_p = 100 v_-$, $s_r = s_p/4$, $T = 100$, $N = 1000$, $D = 0.05$ in case of activation (red) and $D = 0.95$ in case of repression (blue). Darker shades indicate greater values of feedback cost (ranging from $a = 1$ to $a = 2$). Gray denotes the distribution when considering the corresponding non-autoregulated control, serving as a baseline for fitness cost normalization. Distributions are visualized with identical maximal heights.

remains $s_r$, for autoregulation it instead increases to $a s_r$ with $a \geq 1$ (Fig. 7a).

To understand how this change impacts the dominant strategy, we first focus on the case when mutations have minimal impact, so that the population consists entirely of binders. As a result, the P-cost is zero for all four schemes. Furthermore, positive and negative control have identical R-cost, both when considering non-autoregulated and autoregulated motifs (Fig. 2c). While autoregulation reduces the time when the R-cost is suffered as demand decreases, non-autoregulated strategies outperform their autoregulated counterparts once feedback becomes too expensive (Fig. 7a). These results hold even when mutations are considered. In particular, data in Supplementary Figs. 32–34 confirm that (i) the pattern of how the use-it-or-lose-it and the wear-and-tear principles emerge behind the dominant strategies remains largely unaffected, (ii) non-autoregulated strategies can outperform their autoregulated counterparts as $a$ increases, and (iii) this transition happens at lower values of $a$ as the demand increases.

As the R-cost for autoregulation increases with the duration of the induced phase $T_i = DT$ (Fig. 2c), we expect that the sensitivity of the dominant autoregulated strategy to changes in $a$ also increases with the demand $D$, confirmed in Fig. 7b. In particular, while the fitness cost of self-activation increases only slightly when compared to its non-autoregulated counterpart in the low demand case, performance of self-repression in the high demand limit quickly degrades. Furthermore, the results in Fig. 7b echo our finding in Fig. 5 about increasing the R-cost (this time via $a$ instead of $s_r$), giving rise to a differential impact on population composition: more stringent selection against non-binders when relying on self-repression, and the spread of loss-of-function mutations in case of self-activation. This also reveals a trade-off between the fraction of non-binders $x$ and the margin in $a$ for preserving the dominant strategy. For instance, when it is underpinned by the wear-and-tear principle, in the low demand case $a$ can be increased substantially and self-activation would still emerge as dominant (e.g., it remains only about 15% as expensive as non-autoregulated positive control when $a = 2$ in Fig. 7b), although at the price of increased prevalence of non-binders. Conversely, in the high

demand case negative autoregulation promotes the elimination of deleterious mutations, however, the margin in $a$ is considerably smaller to preserve the dominance of self-repression (e.g., non-autoregulated negative control dominates its autoregulated counterpart once $a > 1.07$ in Fig. 7b).

## Reduced bioenergetic cost may contribute to the prevalence of autoregulation in model organisms

We next turn our attention to the transcriptional regulatory network of *B. subtilis*, *C. glutamicum*, and *E. coli*, model organisms with the most comprehensive data availability[58,59] (Supplementary Fig. 35). Our results suggest that self-activation and self-repression should be prominently featured among regulators, hence in what follows we concentrate on whether the reduced bioenergetic cost of autoregulated control schemes could indeed confer an evolutionary advantage, and thus contribute to their high prevalence[60,61].

To this end, we first note that autoregulation is preferentially selected by evolution when choosing regulators. While the average prevalence of autoregulation is only 27% among TFs (each TF is only counted once), it is instead 35% among all regulators (each TF is counted as many times as it appears as an activator/repressor), with the overrepresentation ranging between 2–16 percentage points (Fig. 8a). This is not surprising, as autoregulated TFs offer a multitude of beneficial properties[62–67]. Importantly, if R-cost reduction was a negligible factor, we would expect identical prevalence among all regulators and among the subset that control gene targets synergistically (Fig. 8b), otherwise the frequency in the latter should be greater (Supplementary Section 7). Thus, for each organism our two samples of interest comprise regulators of genes that are either positively or negatively controlled, focusing on 86%, 50%, and 40% of target genes in *B. subtilis*, *C. glutamicum*, and *E. coli*, respectively (Supplementary Table 2).

Using the frequency of autoregulation among TFs as a baseline, the overrepresentation of this motif is greater among synergistic regulators (Fig. 8c) than among all regulators (Fig. 8a): the mean difference is 25 percentage points, with the overrepresentation ranging

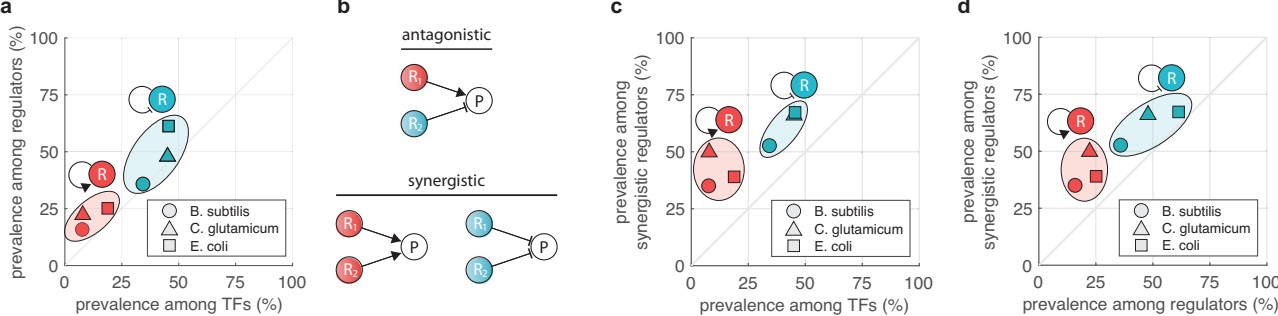

**Fig. 8 | Self-activated/self-repressed TFs are overrepresented among regulators of genes that are synergistically activated/repressed.** Data were obtained by compiling a list of all known regulatory interactions between TFs and gene targets for *B. subtilis*, *C. glutamicum*, and *E. coli* (Supplementary Section 7). **a** Autoregulated motifs are preferentially selected as regulators considering their baseline frequency among TFs. **b** Genes are regulated either synergistically (all TFs are either activators or repressors) or antagonistically (TFs contain both activators and repressors). Potential autoregulation of the TFs is not indicated. **c** The prevalence of autoregulation among synergistic regulators is greater than their baseline frequency among TFs. **d** The prevalence of autoregulation among synergistic regulators is greater than their modified baseline frequency among all regulators.

between 18–42 percentage points (Supplementary Table 2). Therefore, beneficial properties of autoregulation yield their preferential selection for regulators (Fig. 8a), but crucially, the frequency of autoregulation among synergistic regulators further exceeds this modified and elevated baseline by 17 percentage points on average (Fig. 8d), with the overrepresentation ranging between 6–28 percentage points (Supplementary Table 2). To quantify the significance of the differences, we performed standard two-sample location tests across all datasets to compute the probability of the modified baseline (autoregulation among all regulators) and the samples of interest (autoregulation among synergistic regulators) coming from the same distribution (leveraging not only the frequencies but also the sample sizes). The resulting p-values are all smaller than 0.001, suggesting that the differences are statistically significant across all three organisms for both self-activation and self-repression, and that the underlying distributions are likely different.

In summary, autoregulated motifs are overrepresented among synergistic regulators compared to their frequency among all regulators. This suggests that autoregulation likely offers additional beneficial properties in the former case, leading to their even stronger preferential selection. Our results highlight that reduced R-cost offers an appealing explanation as one of the factors that contribute to establishing the prominent role of autoregulation.

## Discussion

Given the prevalence of autoregulation in transcription networks[60,61], it is not surprising that this network motif offers numerous advantages. For instance, self-repression can accelerate temporal responses[62,63] and reduce stochastic fluctuations[64,65], whereas self-activation underpins cellular memory[66] and helps cell populations to maintain a mixed phenotype to assure optimal performance in stochastic environments[67]. As autoregulation ensures that a TF is only expressed when needed, our results highlight that the abundance of this motif may also stem from its reduced bioenergetic and fitness cost relative to non-autoregulated control.

We confirm this both in small and large populations by demonstrating that non-autoregulated regulation practically never dominates its autoregulated counterpart except when the additional cost of feedback is substantial or in the presence of regulatory delays. The dominant strategy is generally in accordance with the wear-and-tear principle, the use-it-or-lose it principle emerges only in a narrow region of population size and timescale of environmental shifts. Our work thus further highlights that demand for a beneficial gene product and whether positive or negative regulation offers superior performance are tightly coupled. These results considerably expand our prior

understanding focusing only on non-autoregulated control[26], and may explain why self-activation and self-repression show strong clustering and preferential localization across functional subsystems (Supplementary Fig. 38) with different temporal demand profiles[68].

As the performance of autoregulation degrades with delay, it is understandable that the transcription network of *E. coli* has an essentially feedforward structure, where feedback occurs primarily in the form of autoregulation[69,70]. Further strengthening the connection between evolutionary selection and autoregulation, the latter most likely emerged as a result of gene duplication[71], a core factor in the origin of mutational robustness[72] resulting in the accumulation of phenotypically cryptic genetic variation[73] promoting evolvability[74]. This link between autoregulation and mutational robustness was confirmed by recent experimental advances, suggesting that regulatory feedback may be an important element of the network architectures that confer mutational robustness across biology[75].

The results presented in this paper focus on the impact of autoregulation by putting the spotlight on fitness advantage, as this factor plays a pivotal role in establishing the dominant strategies that emerge as a result of competition and selection. In synthetic biology applications, however, while the fitness cost can be crucial in certain scenarios, e.g., when studying the cost of plasmid acquisition and maintenance[76], often the rate at which non-binders emerge is instead of primary concern to avoid compromising the genetic stability of engineered synthetic circuits[77]. Importantly, our approach can be applied to characterize how autoregulation impacts the proliferation of deleterious mutations considering timescales and population sizes typical in synthetic biology applications (Supplementary Section 8) ranging from microfluidics experiments to bioreactor-based contexts[78–82]. Crucially, our work reveals that while self-activation facilitates the spread of non-binders, self-repression acts against this phenomenon, and these effects are amplified as the cost of regulation increases. Thus, our results not only uncover how genetic design choices alter population structure, growth dynamics, and evolutionary outcomes, but they can also be leveraged to minimize the prevalence of cells that harbor non-functional genetic modules to design evolutionarily robust synthetic gene circuits[83,84].

Our model-based analysis raises several experimentally testable hypotheses. Does the wear-and-tear principle emerge in small populations and at the critical timescale of environmental shifts in case of sufficiently strong selection? Is it true that while negative autoregulation curbs the emergence of detrimental mutations, self-activation instead facilitates their spread? To test these predictions, all regulatory schemes featured here can be constructed using existing synthetic biology toolkits and parts[85–88], and key variables can be conveniently

tuned to reveal their role in establishing the dominant control scheme. Demand and period length can be varied by creating defined environments to control when the gene product is needed[89,90] using automated cell culture systems[81,82]. Mutation rates can be adjusted by using UV radiation or CRISPR-guided mutagenesis[91,92]. The P-cost can be tuned by modulating selection pressure, for instance, via the composition of the growth media[93–96], whereas the R-cost can be altered through codon (de)optimization[97]. Finally, delay can be modulated leveraging regulatory cascades[57]. By ensuring that experiments with maximal information content are selected[98], it is possible to efficiently test whether the predicted regulatory mechanisms actually emerge as dominant strategies in each environment, thus to promote the design of biosystems that operate robustly under inevitable evolutionary forces[99].

Biology has evolved powerful and creative solutions to control gene expression by selecting the optimal variant(s) among a wide array of competing control mechanisms. Our results reveal how the interplay of biophysical parameters and environmental factors together shape the emergence of dominant regulatory strategies. This can be leveraged both to shed light on evolutionary organizing principles underpinning the transcription networks of living organisms, and also to guide the design of synthetic gene circuits, for instance, when selecting among alternative modes of regulation to implement biomolecular controllers and insulation devices[37–52] to facilitate the modular design of complex synthetic gene circuits.

## Methods
### Parameters
Wild-type *E. coli* grown under optimal conditions typically has mutation rates on the order of $10^{-3}$ mutations per genome per generation[100]. Considering the typical genome size of bacteria, this corresponds to approximately $10^{-9}$–$10^{-8}$ mutations per base per generation[101–103], though this rate may depend on population size[104] and expression levels[105]. Furthermore, hypermutators with up to $10^4$-fold greater mutation rates can occur under laboratory conditions, and more frequently in natural bacterial populations[100]. Assuming roughly 100 sensitive nucleotide positions, we thus estimate the rate of loss-of-function mutations to span the range $v_- \approx 10^{-7}$–$10^{-3}$ per generation. Since gain-of-function mutations are assumed to be less probable, we consider $v_+ = v_-/10$ throughout the paper matching experimental estimates[106]. Selection intensity is notoriously hard to measure[107–110], however, based on estimates for codon bias[111], we consider $s_p/v_- = 10$ and $s_p/v_- = 100$ in case of weak and strong selection, respectively. Finally, we assume that $s_r < s_p$ for a typical target gene, otherwise there would be no evolutionary selection pressure to regulate the expression of the product. Throughout the paper, the period $T$ is measured in number of generations, whereas the mutation rates $v_-$ and $v_+$ as well as the selection intensities $s_p$ and $s_r$ are all given in 1/generation.

### Stochastic simulation of the evolutionary dynamics
The standard Wright-Fisher model assumes the following: discrete and non-overlapping generations with a constant population size $N$, with each member replaced in every generation[53,54]. Introduce $s = s_{nb} - s_b$ where $s_{nb}$ and $s_b$ are the fitness cost incurred by the non-binders and binders, respectively, and let $n_t$ denote number of non-binders in the population in generation $t$ (i.e., $n_t = 0, 1, 2, ..., N$). We first generate the number of gain-of-function and loss-of-function mutations $m_+$ and $m_-$ from Poisson distributions with means $n_t v_+$ and $(N - n_t)v_-$, respectively, so that the fraction of non-binders in the population becomes $x = (n_t + m_- - m_+)/N$. The number of non-binders $n_{t+1}$ in the next generation is drawn from a Binomial distribution with success probability $x' = x - sx(1 - x)/(1 - sx)$ to account for the fitness difference $s$ (thus the number of offsprings produced) between non-binders and binders[26].

### Periodic steady state distribution
Considering the periodic selection pressure in the evolutionary dynamics depicted in Fig. 2, the distribution $P(x, t)$ of $x$ approaches a periodic steady state distribution (subsequent to a transient that depends on the initial condition). This can be estimated for $0 < x < 1$ considering $\frac{\partial P(x,t)}{\partial t} = -\frac{\partial j(x,t)}{\partial x}$ with $j(x,t) = -\frac{1}{2N}\frac{\partial}{\partial x}[x(1-x)P] + [\nu_- - (\nu_+ + \nu_- + s)x + sx^2]P$ using the diffusion approximation[26]. To compute the probabilities $P(0, t)$ and $P(1, t)$ at the boundaries $x = 0$ and $x = 1$, we consider the flux conditions $\frac{dP(0,t)}{dt} = -j(0,t)$ and $\frac{dP(1,t)}{dt} = j(1,t)$. After estimating the steady state distribution of $P(x, t)$ using the algorithm developed in ref. 55 implementing the above steps, the average fitness cost $\bar{s}$ during one period can be computed as in (2) with $x(t)$ replaced by $x'(t) = P(1,t) + \int_0^1 x(t)P(x,t)\,dx$, following the approach in ref. 26.

### Statistical analysis
We perform two-sample location tests comparing the success rate (presence of positive/negative autoregulation) observed in a reference (regulators of genes) and our samples of interest (regulators of synergistically controlled genes). We assume that the underlying distributions are Binomial with $n_r$ and $n$ trials (sample size) and $\theta_r$ and $\theta$ success rates in the reference and in the sample of interest, respectively. Therefore, the number of successes are given by $X_r \sim \text{Binom}(n_r, \theta_r)$ and $X \sim \text{Binom}(n, \theta)$, respectively. With this, $\hat{\theta}_r = X_r/n_r$ and $\hat{\theta} = X/n$ are non-biased estimators of the unknown success rates.

With the null hypothesis $H_0 : \theta = \theta_r$ of identical success rates, we are interested in the probability $p = \mathbf{P}(\Omega \geq \omega \,|\, H_0)$ of observing the value $\omega$ of the test statistic $\Omega$ (number of positively/negatively autoregulated regulators) at least as extreme as if the null hypothesis was true. As $n_r\hat{\theta}_r, n_r(1-\hat{\theta}_r), n\hat{\theta}, n(1-\hat{\theta}) > 5$ for all datasets we consider, from the Central Limit Theorem it follows that

$$\hat{\theta} - \hat{\theta}_r \sim \mathcal{N}\left(\theta - \theta_r, \sqrt{\frac{\theta(1-\theta)}{n} + \frac{\theta_r(1-\theta_r)}{n_r}}\right).$$

Since $\theta_r$ and $\theta$ are unknown, and there are infinitely many choices that satisfy the null hypothesis, we follow the standard choice of using the pooled proportion $\hat{\theta}_0 = (X_r + X)/(n_r + n)$ in place of both, as it satisfies the null hypothesis and it is consistent with our data. With this, we obtain that $p \approx 1 - \Phi(z)$ where $\Phi(\cdot)$ is the cumulative distribution function of the standard normal distribution, together with $z = (\hat{\theta} - \hat{\theta}_r)/\sqrt{\hat{\theta}_0(1-\hat{\theta}_0)(n_r^{-1} + n^{-1})}$.

### Reporting summary
Further information on research design is available in the Nature Portfolio Reporting Summary linked to this article.

## Data availability
CoryneRegNet 7.0 data were downloaded from https://exbio.wzw.tum.de/coryneregnet/processToDownalod.htm[58]. PRODORIC data can be accessed using the API at https://www.prodoric.de/api/[59]. Data on the functional organization of the transcription regulatory network of *E. coli* were downloaded from https://www.pnas.org/doi/full/10.1073/pnas.1702581114[112]. Source data are provided with this paper.

## Code availability
The manuscript does not rely on custom mathematical algorithms or software. Simulation data were generated and analyzed as described in the "Methods" using built-in MATLAB (version R2023a) functions. The MATLAB scripts used to obtain the results featured in the paper are publicly available at https://github.com/qbionet/evolutionary-selection. Additional information is available from the corresponding author upon request.

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

## Acknowledgements

This work was supported by research funds from New York University Abu Dhabi.

## Author contributions

A.G. conceived and designed the research, collected and analyzed the data, prepared the figures, and wrote the manuscript.

## Competing interests

The author declares no competing interests.
