## [Peer Review File · Nature Communications]

Reviewers' Comments:

Reviewer #1:

Remarks to the Author:

The manuscript provides a comparative analysis of different regulatory motifs for gene expression control, comparing autoregulated vs. non-autoregulated expression as well as activation vs. repression, including their impact at the population level in an evolutionary setting. Autoregulation appears to be more favourable, apart from the cases in which either the costs of feedback or regulatory delays are substantial.

The manuscript is well written, the work seems original and the results look interesting and insightful.

A small comment. In the SI Sections 1.1 and 1.2, $K(I)$ and $k(I)$ seem to be functions of I . What is then their functional expression? Does it play a role?

Reviewer #2:

Remarks to the Author:

This paper presents modelling results that explore how evolutionary pressures affect the emergence and stability of different "core" gene regulatory motifs. A model is developed that allows for the costs of regulators and the fitness of a design to be tied to external environmental fluctuations and is used to map out how evolutionary pressures lead to dominant motifs when varying key parameters. The author demonstrates that the model supports several well-established evolutionary theories and precisely describe the scenarios when autoregulation is beneficial and whether positive or negative regulation will dominate. Finally, these findings are explored in the context of gene regulatory networks from a range of model microbes, demonstrating that the energetic cost of expressing a regulator may account for the positive selection of autoregulation.

The paper is exquisitely written, the mathematics and analyses were sound, and it was a joy to read. It is incredibly rare to review such a well-polished manuscript that was further elevated by some excellent figures that very clearly illustrated the core points and summarized the results. Other than a few very minor typos, the only slight weakness I felt was the relatively brief connection with real biological networks at the end. Expanding on this element would be helpful for readers to better see the benefits of such modelling efforts.

Overall, I'm happy to support for publication after the following comments are considered, which are all related to strengthening the works connection to real biological systems.

1. Selection for autoregulation is nicely shown in the paper, however, given this selective pressure is rather weak, where specifically in the regulatory networks is autoregulation round. Are there any functional signals that emerge that might help further support the energetic costs being crucial?
2. The parameters of the model are unitless which enables relative comparisons, but where might natural regulatory networks sit in parameter space given what is known about the energetic costs of gene expression, mutation rates, and rates of selection. It would be nice if something more quantitative could be shown in this regard, although I do appreciate the challenges in doing so.
3. I was disappointed to read that code was only available on request. I would suggest that it is essential **all** modelling studied provide their code in public repositories to aid reproducibility and increase the value of the work to the broader field.

I also spotted a few minor typos that need addressing:

Line 22: "ensure predictable" -> "ensure the predictable"

Line 185: "this, define" -> "this, we define"

Line 276: "2%. to" -> "2% to", there are also several "."s that should be removed on page 18. It would be worth checking throughout to see if there are others.

Line 333: "often times the" -> "often the"

Response to Referees

We thank the Editor and the Reviewers for their helpful and constructive feedback. Addressing these has enabled us to improve our work and to increase the clarity of the presentation of our results. In what follows, we turn to Reviewer comments (highlighted in blue) and detail how we addressed them one-by-one. In addition to the revised manuscript, we have also uploaded a copy indicating all the changes (generated using the `latexdiff` package: added text is blue, discarded text is red).

Reviewer #1

The manuscript provides a comparative analysis of different regulatory motifs for gene expression control, comparing autoregulated vs. non-autoregulated expression as well as activation vs. repression, including their impact at the population level in an evolutionary setting. Autoregulation appears to be more favourable, apart from the cases in which either the costs of feedback or regulatory delays are substantial. The manuscript is well written, the work seems original and the results look interesting and insightful.

We thank the Reviewer for their time spent reviewing our manuscript.

A small comment. In the SI Sections 1.1 and 1.2, $K(I)$ and $k(I)$ seem to be functions of I . What is then their functional expression? Does it play a role?

The functional expression of either $K(I)$ or its de-dimensionalized counterpart $k(I)$ has minimal impact on the simulation results presented in Supplementary Section 1. To illustrate this, consider the case when the dissociation constant $K(I)$ takes the form of a Hill function as $K_a(I) = \bar{K} K_0^n / (K_0^n + I^n)$ and $K_r(I) = \bar{K} I^n / (K_0^n + I^n)$ for activation and repression, respectively, where n is the Hill coefficient. As simulation results in Fig. R1 (similar to those featured in Supplementary Fig. 1–2) reveal, the functional form of $K(I)$ and $k(I)$ has negligible impact on the population dynamics.

Furthermore, it is important to note that the results in subsequent sections of the Supplementary Information and in the main text do not depend on the particular modeling choices presented in Supplementary Section 1 (including the functional expression of $K(I)$ and $k(I)$). Instead, they are underpinned by the time-varying selection pressures featured in Fig. 2c. As a result, the particular modeling choices can be flexibly modified without impacting the findings to capture more complex phenomena, for instance, the multimerization of the regulator or the impact of shared cellular resources and the resulting metabolic burden. This is explicitly mentioned in Supplementary Section 1.

Fig. R1. **The functional form of $K(I)$ and $k(I)$ has negligible impact on the population dynamics.** Light and dark red corresponds to non-autoregulated and autoregulated activation, respectively. Light and dark blue corresponds to non-autoregulated and autoregulated repression, respectively. Induced and non-induced phases alternate with a period of $T = 20\gamma$, the former (gray shaded regions) takes up 25% of each period for activation and 75% for repression. Simulation parameters are: $\bar{K} = 1000$, $K_0 = 100$, $\epsilon = 10^{-3}$, $\gamma = 1$, $\lambda_p = 0.3$, $\lambda_r = 0.1$, $I = 0$ during T_{ni} and $I = 1000K_0$ during T_i .

Reviewer #2

This paper presents modelling results that explore how evolutionary pressures affect the emergence and stability of different “core” gene regulatory motifs. A model is developed that allows for the costs of regulators and the fitness of a design to be tied to external environmental fluctuations and is used to map out how evolutionary pressures lead to dominant motifs when varying key parameters. The author demonstrates that the model supports several well-established evolutionary theories and precisely describe the scenarios when autoregulation is beneficial and whether positive or negative regulation will dominate. Finally, these findings are explored in the context of gene regulatory networks from a range of model microbes, demonstrating that the energetic cost of expressing a regulator may account for the positive selection of autoregulation.

The paper is exquisitely written, the mathematics and analyses were sound, and it was a joy to read. It is incredibly rare to review such a well-polished manuscript that was further elevated by some excellent figures that very clearly illustrated the core points and summarized the results. Other than a few very minor typos, the only slight weakness I felt was the relatively brief connection with real biological networks at the end. Expanding on this element would be helpful for readers to better see the benefits of such modelling efforts.

Overall, I'm happy to support for publication after the following comments are considered, which are all related to strengthening the works connection to real biological systems.

We thank the Reviewer for their time spent reviewing our manuscript, and for their constructive feedback and suggestions that helped us improve our results. In what follows, we address the helpful comments that we have received one-by-one.

1. Selection for autoregulation is nicely shown in the paper, however, given this selective pressure is rather weak, where specifically in the regulatory networks is autoregulation round. Are there any functional signals that emerge that might help further support the energetic costs being crucial? We thank the reviewer for pointing out this unexplored research direction. To address it, we focused on the functional modules that were identified in *E. coli* (1) based on transcriptomics data (2) to show that self-activation and self-repression show strong clustering and preferential localization across functional subsystems (mentioned explicitly in the Discussion). Considering the prevalence of autoregulation in these modules that represent core biological functions, there is a negative correlation between the frequency of self-activation and self-repression (Supplementary Fig. 38). Comparing the prevalence of these motifs to the baseline observed across all TFs in the dataset, three of the modules appear to be dominated by self-activation and three by self-repression. This could serve as a promising starting point for further investigating the potential role that the reduced bioenergetic cost of autoregulation may play in contributing to its prevalence. For instance, functional module 3 encapsulates the motility system, and comprises five TFs of which two are self-activated. At the other end of the spectrum, functional module 5 encompasses the metabolism of carbon, amino acids, and nucleotides, and comprises 13 TFs of which ten are self-repressed. Focusing on the demand profile of these autoregulated TFs could thus further clarify the potential relationship between demand for a beneficial gene product and the evolutionary advantage that autoregulation may confer.
2. The parameters of the model are unitless which enables relative comparisons, but where might natural regulatory networks sit in parameter space given what is known about the energetic costs of gene expression, mutation rates, and rates of selection. It would be nice if something more quantitative could be shown in this regard, although I do appreciate the challenges in doing so.

We thank the Reviewer for pointing out this important aspect of the model and also for acknowledging the corresponding challenges. The parameters in our model are in fact included with units as detailed in the Methods (indicated explicitly in the Results). In particular, the period T is measured in number of generations, whereas the mutation rates ν_- and ν_+ as well as the selection intensities s_p and s_r are all given in 1/generation. The values featured throughout the paper are selected considering typical values measured or estimated in *E. coli*, as we detail in the Methods.

3. I was disappointed to read that code was only available on request. I would suggest that it is essential *all* modelling studied provide their code in public repositories to aid reproducibility and increase the value of the work to the broader field.

We thank the reviewer for pointing out this shortcoming. To address this issue, the MATLAB scripts that we relied on to obtain the results featured in the paper are made publicly available at <https://github.com/qbionet/evolutionary-selection>.

4. I also spotted a few minor typos that need addressing:

- Line 22: “ensure predictable” → “ensure the predictable”
This typo has been fixed.
- Line 185: “this, define” → “this, we define”
This typo has been fixed.
- Line 276: “2%. to” → “2% to”, there are also several “.”s that should be removed on page 18. It would be worth checking throughout to see if there are others.
The indicated quantities are percentage points (i.e., the unit for the arithmetic difference between two percentages). To avoid any confusion and to increase clarity, “%.” has been changed to “percentage points” throughout the paper.
- Line 333: “often times the” → “often the”
This typo has been fixed.

References

1. Fang, X. *et al.* Global transcriptional regulatory network for Escherichia coli robustly connects gene expression to transcription factor activities. *Proceedings of the National Academy of Sciences* **114**, 10286–10291 (2017).
2. Carrera, J. *et al.* An integrative, multi-scale, genome-wide model reveals the phenotypic landscape of Escherichia coli. *Molecular Systems Biology* **10**, 735–735 (2014).